# Quantitative comparison of lip melanin concentration between smokers and non-smokers using hyperspectral imaging: A Pilot Study

**Jaehun Kwon**[1]                                            kwon8737@sch.ac.kr
**Juhyun Kim**[2]                                           jjjuhyun4502@gmail.com
**Onseok Lee**[*1,2]                                              leeos@sch.ac.kr
[1] *Dept. of Medical IT Engineering, Soonchunhyang University, Republic of Korea*
[2] *Dept. of Software Convergence, Graduate School, Soonchunhyang University, Republic of Korea*

## Abstract

This study analyzed the differences in lip melanin pigmentation between smokers and non-smokers using hyperspectral imaging (HSI) absorbance-based spectral unmixing. The melanin index was derived by separating the melanin components from the lip tissue, where hemoglobin interference was significant. Experimental results confirmed a significant difference in melanin absorbance between smokers and non-smokers, verifying the model was verified. Hyperspectral reflectance images were acquired using a Hyperspectral camera over 470–900 nm (150 bands), and the spectra were converted to absorbance for Beer–Lambert-based analysis. Component concentrations were estimated using non-negative least squares (NNLS) with literature-based extinction coefficients of melanin, deoxyhemoglobin ($Hb$), and oxyhemoglobin ($HbO_2$). The reconstructed spectrum showed high agreement with the original lip absorbance ($R^2 = 0.9496$), supporting stable component separation under hemoglobin interference. In the 650–700 nm band, the smoker group showed higher mean melanin absorbance than the non-smoker group (0.3422 and 0.2538, approximately 1.35-fold, $p = 0.008$).

**Keywords:** Hyperspectral imaging, Spectral unmixing, Lip melanin quantification.

## 1. Introduction

Lip melanin pigmentation caused by smoking is an indicator of cumulative smoking exposure and periodontal health status (Kamma et al., 2007), including probing depth and attachment loss (Multani, 2013). Colorimeters and other similar devices have been used for this evaluation (Martin-Zaman et al., 2023). However, because of hemoglobin signal interference in the lips, which are rich in blood vessels, selectively quantifying melanin using only conventional colorimetric methods is difficult (Vergnaud et al., 2024). Therefore, this study employed HSI-based unmixing to isolate melanin and quantitatively compare concentrations between smokers and non-smokers. In lip tissue, vascular spectral contributions can obscure melanin-specific variation and reduce the reliability of direct color-based interpretation (Jacques, 2013). Therefore, a component-resolved strategy is required to extract melanin-related signals exclusively by excluding hemoglobin-driven effects (Calin et al., 2023). The objective of this study is to establish an objective and component-resolved approach and to compare melanin-related lip absorbance between smokers and non-smokers.

---

[*] Contributed equally

## 2. Methods

This study was conducted on adult participants, comprising non-smokers with no smoking experience and smokers with a smoking history of more than 5 years (approximately 1.75 pack-years). Hyperspectral reflectance images of the lips were acquired using a SNAP-SCAN VNIR camera (Imec, Leuven, Belgium) at 470–900 nm (150 bands). The lips of all the participants were disinfected and dried, and images were captured under identical camera and lighting conditions. The acquired reflectance data were converted to an absorbance vector (Jacques, 2013) according to Eq. 1. Subsequently, non-negative least squares (NNLS)-based linear unmixing was performed to estimate the concentration of each component using power-law-based melanin (Jacques, 2013) and hemoglobin extinction coefficients (Takatani and Graham, 1979). Component coefficients were estimated by the non-negative optimization in (2), and the reconstructed absorbance spectrum was obtained using (3) for comparison with the observed spectrum. The melanin absorbance band for comparison between the smoker and non-smoker groups was set to 650–700 nm, where hemoglobin interference was minimized (Jacques, 2013), and statistical significance was evaluated using an independent samples t-test.

$$\mathbf{y} = -\log_{10}\bigl(R(\lambda)\bigr), \tag{1}$$

where $\mathbf{y} \in \mathbb{R}^L$ denotes the observed absorbance vector across $L$ wavelengths. Component concentrations were estimated using NNLS-based linear unmixing with melanin, deoxyhemoglobin ($Hb$), and oxyhemoglobin ($HbO_2$) extinction bases:

$$\hat{\mathbf{x}} = \arg\min_{\mathbf{x} \geq 0} \left\| \mathbf{y} - \left( \mathbf{E}\mathbf{c} + c_0\mathbf{1} + c_1\tilde{\boldsymbol{\lambda}} \right) \right\|_2^2, \quad \mathbf{E} = \left[ \boldsymbol{\varepsilon}_{\text{mel}}^{(n)}, \boldsymbol{\varepsilon}_{\text{Hb}}^{(n)}, \boldsymbol{\varepsilon}_{\text{HbO}_2}^{(n)} \right], \tag{2}$$

where $\mathbf{c} = [c_{\text{mel}}, c_{\text{Hb}}, c_{\text{HbO}_2}]^\top$, $\mathbf{x} = [\mathbf{c}^\top, c_0, c_1]^\top$, and $\tilde{\boldsymbol{\lambda}}$ is the normalized wavelength vector.

The reconstructed absorbance spectrum is:

$$\hat{\mathbf{y}} = \mathbf{E}\hat{\mathbf{c}} + \hat{c}_0\mathbf{1} + \hat{c}_1\tilde{\boldsymbol{\lambda}}. \tag{3}$$

## 3. Results

The mixed spectrum reconstructed from the estimated separated spectral components exhibited high consistency with the original lip absorbance, with $R^2 = 0.9496$, thereby confirming that the proposed method stably separates melanin components even with hemoglobin interference. According to the box plot analysis, the mean melanin absorbance of the smoker group was 0.3422, approximately 1.35-fold higher than that of the non-smoker group (0.2538, at 650–700nm, $p = 0.008$), as shown in Figure 1. To further validate the reliability of the mixed spectrum reconstruction model as an ablation study, sequential fitting results for a representative smoker sample are presented in Figure 2 as an ablation study. As components were added stepwise, the RMSE decreased from 0.1531 (Model 1: melanin only) to 0.0604 (Model 2: $+Hb$), 0.0593 (Model 3: $+HbO_2$), and 0.0567 (Model 4: full model), yielding an overall improvement of approximately 63.0% relative to Model 1. These results support that adding hemoglobin-related components and baseline terms improves spectral reconstruction in lip spectra.

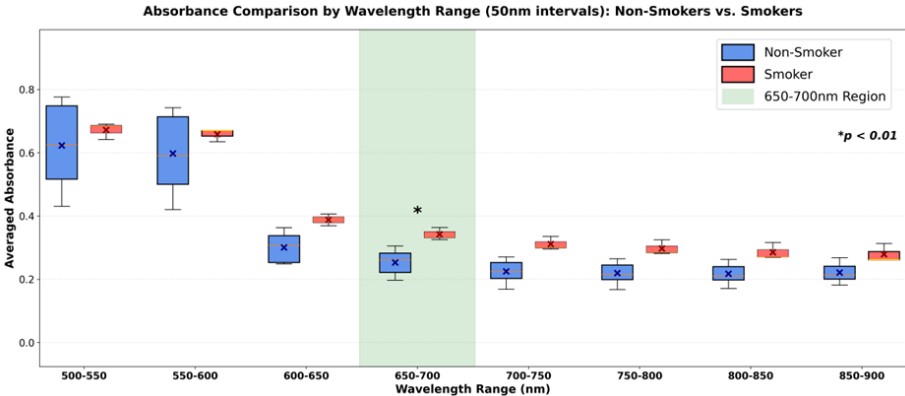

Figure 1: Absorbance Comparison by Wavelength Range: Non-Smokers vs. Smokers. The shaded area (650-700nm) indicates a significant difference ($p < 0.01$).

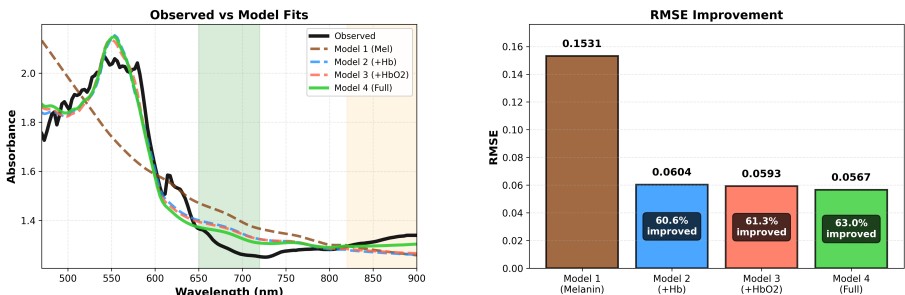

Figure 2: Sequential spectral fitting in a representative smoker sample: stepwise inclusion of components improved reconstruction agreement.

## 4. Discussion & conclusion

This study demonstrates that absorbance-based hyperspectral unmixing enables objective quantification of melanin-specific spectral components in lip tissue, confirming a significant difference between smokers and non-smokers. The framework's physical validity is supported by high spectral reconstruction agreement ($R^2 = 0.9496$). Smokers showed significantly higher melanin absorbance at 650–700 nm ($p = 0.008$), confirming clinical relevance for periodontal health assessment, with potential for future applications in cosmetic dermatology.

## Acknowledgments

This research was supported by the MSIT(Ministry of Science, ICT), Korea, under the National Program for Excellence in SW, supervised by the IITP(Institute of Information & communications Technology Planning & Evaluation) in 2026(2021-0-01399).

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
