# OpenReview forum: "Quantitative comparison of lip melanin concentration between smokers and non-smokers using hyperspectral imaging: A Pilot Study"
_MIDL.io/2026/Short_Papers — MIDL 2026 - Short Papers Poster_

### Official Review · Reviewer_ueSK · 2026-05-06
**Solid physics-driven HSI analysis for smoking-related pigmentation, but limited ML novelty**

**Rating:** 3
**Confidence:** 5

**Review:**

- This is a well-motivated and physically grounded study that applies hyperspectral imaging to a clinically relevant problem: objective quantification of lip pigmentation differences related to smoking. The use of spectral unmixing with known extinction coefficients is appropriate and aligns with established biomedical optics principles.

- The main strength of the paper is its clear connection to physical modeling (Beer–Lambert law, NNLS unmixing) rather than purely data-driven learning. The pipeline is simple but interpretable, and the reconstruction accuracy (R² ≈ 0.95) suggests the model is capturing meaningful spectral structure. The experimental setup is also reasonably controlled, and the statistical comparison between smokers and non-smokers is clearly presented.

- However, from a machine learning and MIDL perspective, the novelty is limited. The method is essentially a classical linear spectral unmixing pipeline applied to a new dataset, rather than a new modeling or learning contribution. There is no learning-based component beyond optimization under constraints, and no comparison with modern HSI or representation learning approaches.

- Another limitation is the small scale and pilot nature of the study. While statistically significant differences are reported, the dataset size and demographic variability are not clearly described, which raises questions about generalizability. Additionally, the reliance on fixed extinction coefficients from literature may introduce bias if inter-subject variability is high.

- The ablation-style spectral reconstruction analysis is a nice addition, but it mainly validates the expected behavior of the linear model rather than demonstrating a novel insight. More importantly, there is no comparison against alternative methods (e.g., PCA-based unmixing, nonlinear models, or learning-based spectral decomposition), which would help position the contribution more strongly.

- Clarity is generally good, and the physical formulation is clearly explained, but the paper reads more like an applied biomedical optics study than a deep learning contribution.

-  this is a solid and well-executed pilot study with clear clinical relevance, but limited methodological novelty from a deep learning perspective.

**Summary:**

This paper presents a hyperspectral imaging (HSI) based method to quantify lip melanin differences between smokers and non-smokers. The approach uses absorbance conversion and non-negative least squares (NNLS) spectral unmixing to separate melanin from hemoglobin components (Hb and HbO2). A hyperspectral dataset (470–900 nm, 150 bands) is collected under controlled conditions, and melanin concentration is estimated using literature-based extinction coefficients. Results show a statistically significant difference in melanin absorbance between smokers and non-smokers in the 650–700 nm range (p = 0.008), with smokers showing higher values. The reconstructed spectra achieve high agreement (R² ≈ 0.95), supporting the validity of the unmixing model.

**Strengths:**

- Strong physical grounding using Beer–Lambert law and NNLS spectral unmixing
- Clear and clinically relevant application (smoking-related pigmentation analysis)
- High spectral reconstruction quality (R² ≈ 0.95), indicating stable decomposition
- Controlled acquisition setup with standardized imaging conditions
- Statistically significant findings between groups (p = 0.008)
- Simple, interpretable pipeline suitable for biomedical use

**Weaknesses:**

- Limited ML novelty; primarily a classical spectral unmixing pipeline, and no comparison with alternative methods (linear/nonlinear or learning-based approaches)
- Small-scale pilot study with limited discussion of dataset size and variability, and also heavy reliance on fixed literature-based extinction coefficients
- Ablation study mainly confirms expected model behavior rather than new insight

**Justification Of Rating:**

As described above sections.

---

### Decision · Program_Chairs · 2026-05-08

Accept (Poster)